# HIF-α Activation Impacts Macrophage Function during Murine *Leishmania major* Infection

**DOI:** 10.3390/pathogens10121584

**Published:** 2021-12-06

**Authors:** Manjunath Bettadapura, Hayden Roys, Anne Bowlin, Gopinath Venugopal, Charity L. Washam, Lucy Fry, Steven Murdock, Humphrey Wanjala, Stephanie D. Byrum, Tiffany Weinkopff

**Affiliations:** 1Department of Microbiology and Immunology, College of Medicine, University of Arkansas for Medical Sciences, Little Rock, AR 72205, USA; mnbettadapur@ualr.edu (M.B.); hroys@uams.edu (H.R.); abowlin@uams.edu (A.B.); gvenugopal@uams.edu (G.V.); lfry@uams.edu (L.F.); sjmurdock@uams.edu (S.M.); hmwanjala@ualr.edu (H.W.); 2Department of Biochemistry and Molecular Biology, College of Medicine, University of Arkansas for Medical Sciences, Little Rock, AR 72205, USA; cwasham@uams.edu (C.L.W.); sbyrum@uams.edu (S.D.B.); 3Arkansas Children’s Research Institute, Little Rock, AR 72202, USA

**Keywords:** *Leishmania*, leishmaniasis, macrophages, HIF-α

## Abstract

Leishmanial skin lesions are characterized by inflammatory hypoxia alongside the activation of hypoxia-inducible factors, HIF-1α and HIF-2α, and subsequent expression of the HIF-α target VEGF-A during *Leishmania major* infection. However, the factors responsible for HIF-α activation are not known. We hypothesize that hypoxia and proinflammatory stimuli contribute to HIF-α activation during infection. RNA-Seq of leishmanial lesions revealed that transcripts associated with HIF-1α signaling were induced. To determine whether hypoxia contributes to HIF-α activation, we followed the fate of myeloid cells infiltrating from the blood and into hypoxic lesions. Recruited myeloid cells experienced hypoxia when they entered inflamed lesions, and the length of time in lesions increased their hypoxic signature. To determine whether proinflammatory stimuli in the inflamed tissue can also influence HIF-α activation, we subjected macrophages to various proinflammatory stimuli and measured VEGF-A. While parasites alone did not induce VEGF-A, and proinflammatory stimuli only modestly induced VEGF-A, HIF-α stabilization increased VEGF-A during infection. HIF-α stabilization did not impact parasite entry, growth, or killing. Conversely, the absence of ARNT/HIF-α signaling enhanced parasite internalization. Altogether, these findings suggest that HIF-α is active during infection, and while macrophage HIF-α activation promotes lymphatic remodeling through VEGF-A production, HIF-α activation does not impact parasite internalization or control.

## 1. Introduction

Leishmaniasis is an inflammatory disease caused by vector-borne, obligate intracellular protozoan parasites of the genus *Leishmania*. Leishmaniasis is endemic in many developing countries in tropical and subtropical regions of the globe, where up to 1.7 million new cases in 98 countries occur annually [1,2]. *Leishmania* manifests in three forms: (1) visceral leishmaniasis (VL), which is systemic and fatal if untreated, (2) cutaneous leishmaniasis (CL), which causes nodules, papules, and lesions on the surface of the skin, and (3) mucocutaneous leishmaniasis (MCL), wherein parasites from primary cutaneous lesions spread to other parts of the body [3]. Infections with different species of *Leishmania* manifest in different clinical forms. In particular, *Leishmania major* is responsible for CL in the Eastern Mediterranean and North Africa, where the disease is considered a major public health problem and accounts for 70% of CL cases worldwide [4]. The burden on these countries, already high due to current socioeconomic conditions, limited resources and medical infrastructure, and civil strife, is further exacerbated by the lack of a vaccine and ineffective chemotherapeutic treatment for the disease [3,4].

The transcription factor hypoxia-inducible factor-1α (HIF-1α), which is involved in cellular stress and the response to decreased oxygen availability, is present in lesions from humans and mice infected with *Leishmania* parasites [5,6,7,8]. The disease severity is also associated with enhanced levels of HIF-1α in human and murine CL [6,8]. We previously reported that target genes of HIF-1α and HIF-2α are elevated following *L. major* infection in vivo [6,8,9]. These data suggest that both HIF-1α and HIF-2α are active in the skin during CL. Moreover, HIF-1α and the downstream target gene VEGF-A are elevated in infected and uninfected macrophages in the lesions [8]. Although other species of *Leishmania* parasites activate HIF-1α [10,11], *L. major* parasites alone do not induce HIF-1α accumulation or target gene expression in macrophages [6]. In contrast, infected macrophages cultured under hypoxic conditions or stimulated with proinflammatory signals such as IFNγ and LPS enhanced HIF-1α activation [6]. HIF-1α activated by these inflammatory signals promoted leishmanicidal macrophage activity, but HIF-1α stabilization alone in the absence of inflammatory signals did not promote parasite killing [6]. Altogether, these data suggest that the tissue microenvironment, rather than the parasites, drives pan-HIF-α activation during *L. major* infection.

Macrophages perform multiple functions in the skin following *L. major* infection. Macrophages play an essential role in antileishmanial immunity by recognizing, phagocytosing, and killing parasites [3]. In addition to phagocytosing parasites, dermal macrophages also phagocytose apoptotic cells and debris as part of the wound healing response [12,13]. Although macrophages present antigens in other infections, *Leishmania* infection in macrophages impairs antigen presentation and IL-12 release, thereby dampening CD4^+^ Th1 immune responses [14,15,16,17]. During an efficient antileishmanial immune response, CD4^+^ Th1 cells produce IFNγ, which signals to macrophages to kill parasites in an NO- and ROS-dependent manner, which ultimately leads to parasite control and lesion resolution [3]. While the importance of macrophages in the immune response has been well-characterized, recent evidence has shown that macrophages also play a role in vascular remodeling at the site of infection [8]. Macrophages produce VEGF-A, which binds to VEGFR-2 on lymphatic endothelial cells to induce lymphangiogenesis [7,8]. Macrophage VEGF-A is dependent on HIF-α signaling, and mice deficient in myeloid pan-HIF-α signaling exhibit increased pathology [8]. Therefore, in addition to their well-characterized role in immune responses and parasite killing, macrophages also orchestrate the expansion of the lymphatics, which leads to lesion resolution during *L. major* infection.

Given that HIF-1α and HIF-2α are activated in leishmanial lesions [6,7,8,9], and macrophage HIF-α activation contributes to lesion resolution [8], we investigated factors contributing to HIF-α activation as well as other downstream consequences of HIF-α activation during *L. major* infection. We developed an in vitro system where HIF-α is constitutively active by exposing macrophages to dimethyloxalylglycine (DMOG), a chemical agent that inhibits prolyl-4-hydroxylase domain (PHD) enzymes that degrade HIF-α isoforms, thereby stabilizing HIF-α over the course of infection [18]. Therefore, macrophage DMOG treatment mimics the HIF-1α and HIF-2α activation detected in leishmanial lesions. Using this system, we found that HIF-α activation does not impact parasite entry or control, but rather that basal HIF-α signaling restricts parasite internalization by macrophages.

## 2. Results

### 2.1. The HIF-1α Signaling Pathway Is Increased during L. major Infection In Vivo

Leishmanial lesions in the skin are characterized by inflammatory hypoxia. To determine if hypoxia at the site of infection is associated with increased HIF-α signaling during *L. major* infection in the murine model, we infected C57BL/6 mice with *L. major* parasites. At 4 weeks p.i., when mice were presented with lesions, we performed RNA-Seq on infected and control uninfected ears of naïve mice. Differentially expressed genes (DEGs) between infected and naïve ears were identified by limma voomWithQualityWeights in R, and pathway enrichment analysis of DEGs was performed using EGSEA against the KEGG database. The gene expression profiles derived from the RNA-Seq data were considered statistically significant if fold change >2 and *p* < 0.05. Hierarchical clustering analysis revealed that many transcripts that were significantly induced following *L. major* infection are associated with the HIF-1α signaling pathway, including *Egln1 (Phd2), Egln3, (Phd3), Eno2, Hif1a, Hk2, Hk3, Ldha, Nos2, Vegfa*, and *Vhl* (Figure 1). Altogether, our global transcriptomic analysis suggests that HIF-1α signaling is increased during *L. major* infection in vivo, consistent with findings that leishmanial lesions are hypoxic.

### 2.2. Myeloid Cells Experience Inflammatory Hypoxia in Leishmanial Lesions

We hypothesize that inflammatory hypoxia further potentiates chronic inflammation in leishmanial lesions. We previously reported that inflammatory monocytes, which are massively recruited to the site of infection following *L. major* inoculation, are less hypoxic at the cellular level compared to macrophages in the skin during infection [9]. However, it was not clear whether this was a cell-intrinsic feature that distinguishes monocytes from macrophages, or if the tissue microenvironment imprints a hypoxic signature upon these cells. Therefore, we utilized an adoptive transfer system in which CD11b^+^ cells isolated from the bone marrow and spleens of CD45.1 donor mice were injected into CD45.2 recipient mice that were infected 4 weeks prior to the cell transfer with *L. major* parasites. Infected mice were then treated with pimonidazole to determine which cells experience hypoxic conditions at the cellular level (Figure 2A). In most studies, hypoxia is detected at the tissue level; however, this adoptive transfer model examines hypoxia at a higher resolution and determines if monocytes can become hypoxic as they enter inflamed tissues at the cellular level. Using this system, coupled with flow cytometry, we found that CD45.1^+^CD11b^+^ cells were detectable in the lesions of infected CD45.2^+^ mice, suggesting that myeloid cells are actively recruited to the site of infection (Figure 2B). Further analysis of the transferred myeloid cells from the donor mouse present at the site of infection showed that CD45.1^+^ cells displayed a CD11b^+^Ly6G^+^ neutrophil or CD11b^+^Ly6G^−^CD64^+^ macrophage phenotype. Confirming previous results, these data suggest that monocytes differentiate into macrophages following *Leishmania* infection [19,20,21]. The amount of hypoxia staining based on the pimonidazole median fluorescence intensity (MFI) for each myeloid cell type from donor CD45.1^+^ cells as well as the endogenous host CD45.2^+^ cells was investigated (Figure 2C). These studies revealed that CD11b^+^Ly6G^+^ neutrophils exhibited lower levels of pimonidazole staining compared to macrophages in both the CD45.2^+^ host and CD45.1^+^ transferred cells. Importantly, macrophages derived from CD45.1^+^ donor mice possessed similar pimonidazole MFIs compared to endogenous CD45.2^+^ macrophages (Figure 2D). These data suggest that monocyte-derived macrophages recruited from the circulation experience hypoxic conditions in inflamed lesions, which may impact their metabolic reprogramming and thus the differentiation or function of these cells.

### 2.3. The Length of Time a Myeloid Cell Spends in Leishmanial Lesions Dictates the Hypoxic Signature of That Cell

Following *L. major* infection, monocytes migrate into lesions and exhibit a hypoxic signature, but the length of time in the inflamed tissue may influence the hypoxic signature of the cell, further impacting myeloid cell differentiation and function. To determine if increased time in the lesion enhances hypoxia at the cellular level, we investigated myeloid cells at different stages of their transition from monocytes to macrophages based on the downregulation of Ly6C (an inflammatory monocyte marker) and upregulation of CD64 (a macrophage marker) (Figure 3A). This strategy of examining myeloid cells that are actively differentiating from Ly6C^hi^CD64^−^ inflammatory monocytes to Ly6C^−^CD64^+^ macrophages, coupled with pimonidazole staining, determined that hypoxia occurred at the cellular level across the stages of transition. Confirming previous work [9], we found that Ly6C^hi^CD64^−^ inflammatory monocytes exhibited less pimonidazole staining than Ly6C^−^CD64^+^ fully differentiated macrophages (Figure 3B,C). Moreover, as monocytes transitioned to Ly6C^hi^CD64^+^ cells, their pimonidazole stain positivity also increased. The trend continued, where Ly6C^lo^CD64^+^ cells had higher levels of pimonidazole compared to Ly6C^hi^CD64^+^ cells that eventually peaked in the Ly6C^−^CD64^+^ macrophages, which had the greatest pimonidazole MFI (Figure 3B,C). These data suggest that inflammatory monocytes that enter the inflamed tissue experience hypoxic conditions, and the length of time in the lesion imprints a hypoxic signature upon these cells as they differentiate into macrophages.

### 2.4. Proinflammatory Stimuli and HIF-α Stabilization Induce Macrophage VEGF-A Production in an ARNT/HIF-Dependent Manner during L. major Infection

To determine whether additional proinflammatory stimuli present at the site of infection could also activate HIF-α in addition to hypoxia during infection, we established an in vitro culture system using BMDMs. Macrophages were cultured alone or with a panel of proinflammatory stimuli, including LPS and IFNγ, in the presence or absence of DMOG to stabilize HIF-1α and HIF-2α. These conditions in conjunction with *L. major* infection in macrophages mimic the hypoxic environment and the elevated HIF-1α and HIF-2α expression present at the site of infection in leishmanial lesions [9]. Specifically, macrophages were first infected (or not) with *L. major* parasites (5:1 MOI) and then cultured with LPS alone, IFNγ alone, or both LPS and IFNγ. Compared to uninfected control macrophages, infection with *L. major* parasites alone did not induce VEGF-A production after 24 h (Figure 4A). Amongst uninfected macrophages, only the combination of LPS/IFNγ resulted in significant VEGF-A production (Figure 4A). In infected macrophages, the addition of LPS or LPS/IFNγ led to significant VEGF-A production (Figure 4A). Importantly, VEGF-A production was significantly elevated in all conditions with the addition of DMOG when comparing the same condition with and without DMOG (Figure 4A).

To confirm that DMOG stabilizes HIF-α, leading to HIF-α activation during infection, VEGF-A production was measured from BMDMs generated from LysM^Cre^ARNT^f/f^, which are deficient in myeloid ARNT/HIF-α signaling, and LysM^Cre^ARNT^f/+^ control mice, which exhibit intact ARNT/HIF-α signaling. Using these genetic mouse models demonstrates that DMOG treatment directly affects ARNT/HIF-α signaling and does not act as an off-target effect. Compared to macrophages from LysM^Cre^ARNT^f/+^ mice, LysM^Cre^ARNT^f/f^ macrophages produced significantly less basal VEGF-A (Figure 4A,B). In LysM^Cre^ARNT^f/f^ macrophages, only LPS plus parasites led to significant VEGF-A production that did not increase with DMOG, suggesting that some VEGF-A production during infection can be independent of ARNT/HIF-α signaling (Figure 4B). Importantly, LysM^Cre^ARNT^f/f^ macrophages produced significantly less VEGF-A compared to LysM^Cre^ARNT^f/+^ macrophages in the presence of DMOG for all conditions (Figure 4A,B). Altogether, these results suggest that some proinflammatory stimuli present in leishmanial lesions could activate HIF-α, but the overall effects of the proinflammatory stimuli tested are modest compared to chemical stabilization of HIF-1α and HIF-2α.

### 2.5. DMOG Induces HIF-1α and HIF-2α Activation during L. major Infection

During *L. major* infection in vivo, HIF-1α and HIF-2α, as well as multiple HIF-1α and HIF-2α targets, including *Vegfa*, *Nos2*, and *Arg1*, are elevated at the site of infection [7,8,9]. To determine the effects of pan-HIF-α stabilization during *L. major* infection, we stimulated BMDMs in the presence of DMOG to mimic HIF-1α and HIF-2α stabilization in vitro. Macrophages were generated from LysM^Cre^ARNT^f/f^ and LysM^Cre^ARNT^f/+^ mice to determine the requirement for myeloid HIF-α signaling in response to DMOG administration. Macrophages were exposed to 0.1 or 0.2 mM DMOG prior to infection, and the expression of VEGF-A, which can result from HIF-1α or HIF-2α activation, was examined by real-time PCR. Our results show that *L. major* parasites alone do not induce VEGF-A expression in LysM^Cre^ARNT^f/+^ macrophages (Figure 5A), confirming previous findings [9]. However, DMOG exposure in infected and uninfected macrophages upregulated *Vegfa*, confirming the results in Figure 4. Importantly, VEGF-A expression was lower in LysM^Cre^ARNT^f/f^ macrophages compared to LysM^Cre^ARNT^f/+^ control BMDMs during *L. major* infection in response to DMOG, suggesting that HIF-α mediates *Vegfa* expression during infection. In addition, HIF-1α-specific and HIF-2α-specific target genes were examined with and without infection in BMDMs treated (or not) with DMOG. As with *Vegfa,* the expression of HIF-1α-specific targets *Nos2* and *Pgk1* was significantly higher in uninfected DMOG-treated LysM^Cre^ARNT^f/+^ macrophages compared to uninfected untreated LysM^Cre^ARNT^f/+^ controls, but the same trend was not seen in LysM^Cre^ARNT^f/f^ macrophages (Figure 5B,C). While DMOG increased both HIF-1α-specific targets *Nos2* and *Pgk1* in LysM^Cre^ARNT^f/+^ macrophages compared to untreated cells during infection, this effect was only significant for *Nos2* expression (Figure 5B,C). Importantly, there was no significant difference detected in HIF-1α targets *Nos2* and *Pgk1* in LysM^Cre^ARNT^f/f^ macrophages with DMOG treatment compared to untreated cells (Figure 5B,C). The expression of the HIF-2α-specific target *Arg1* was significantly higher in both uninfected and infected LysM^Cre^ARNT^f/+^ macrophages with DMOG treatment compared to untreated LysM^Cre^ARNT^f/+^ macrophages (Figure 5D). While *Arg1* was not significantly elevated in LysM^Cre^ARNT^f/f^ macrophages with DMOG treatment, some increased *Arg1* was detected in uninfected LysM^Cre^ARNT^f/f^ macrophages, suggesting that some DMOG-induced *Arg1* is independent of ARNT/HIF-α signaling (Figure 5D). The expression of the HIF-2α-specific target *Epo* was higher in both uninfected and infected LysM^Cre^ARNT^f/+^ macrophages with DMOG treatment compared to untreated LysM^Cre^ARNT^f/+^ macrophages, but this result was not significant (Figure 5E). In summary, the findings from this figure show: (1) *Vegfa* and both HIF-1α- and HIF-2α-specific target genes are increased with DMOG treatment, and (2) the infection status of the macrophages does not influence HIF-α activation in response to DMOG, suggesting that the parasite does not directly play a role in macrophage HIF-α activation and, when HIF-α is activated, the parasite does not modulate that response. Altogether, these data indicate that DMOG is an effective tool to simulate *Vegfa* and other HIF-α targets during *L. major* infection.

### 2.6. Macrophage HIF-α Deletion and Stabilization during L. major Internalization and Killing

Given that DMOG simultaneously induces transcripts involved in parasite persistence such as *Arg1*, indicative of an M2 macrophage, and parasite killing such as *Nos2*, indicative of an M1 proinflammatory macrophage, we analyzed parasite persistence in macrophages by monitoring parasite internalization as well as their growth and survival in macrophages with and without DMOG treatment. To determine if HIF-α stabilization impacts the ability of macrophages to phagocytose *L. major* parasites, C57BL/6 BMDMs were treated with DMOG prior to infection. We found an equal number of parasites per macrophage after 2 h of infection in DMOG-treated cells compared to untreated macrophages, suggesting that HIF-α stabilization does not impact macrophage phagocytosis of *L. major* parasites (Figure 6A). To determine if HIF-α stabilization impacts parasite killing, BMDMs were infected and then treated with DMOG for the duration of infection. As we previously reported, DMOG treatment does not impact parasite growth in macrophages for the first 72 h of infection (Figure 6B). Although LPS and IFNγ lead to parasite killing, resulting in lower numbers of parasites after 72 h, DMOG treatment does not impact the effects of LPS and IFNγ (Figure 6B).

While HIF-α stabilization did not affect host–parasite interactions in the first 72 h, we wanted to determine if the absence of HIF-α signaling affects host–parasite interactions early during infection. BMDMs from LysM^Cre^ARNT^f/f^ and LysM^Cre^ARNT^f/+^ mice were infected, and internalized *L. major* parasites were quantified after 2 h. Surprisingly, LysM^Cre^ARNT^f/f^ macrophages possessed a significantly higher number of parasites per macrophage compared to LysM^Cre^ARNT^f/+^ macrophages, independent of DMOG treatment (Figure 6C). These data show that macrophages deficient in ARNT/HIF-α signaling exhibit higher numbers of phagocytosed parasites at 2 h. After 72 h p.i., parasite numbers were higher in LysM^Cre^ARNT^f/f^ macrophages compared to LysM^Cre^ARNT^f/+^ controls, irrespective of DMOG treatment (Figure 6D). LysM^Cre^ARNT^f/f^ macrophages exposed to LPS and IFNγ resulted in lower numbers of parasites when compared to untreated LysM^Cre^ARNT^f/f^ macrophages, suggesting that macrophages deficient in ARNT/HIF-α signaling can control parasites upon proinflammatory stimulation (Figure 6D). However, LysM^Cre^ARNT^f/f^ macrophages possessed a higher number of parasites per macrophage compared to LysM^Cre^ARNT^f/+^ controls following LPS and IFNγ stimulation, indicating that LysM^Cre^ARNT^f/f^ macrophages do exhibit a slight, yet significant, defect in their ability to kill parasites (Figure 6D, significant by *t*-test but not depicted on the graph). Taken together, our findings show that macrophage HIF-α stabilization does not impact parasite phagocytosis or killing, but ARNT/HIF-α signaling restricts parasite entry into macrophages.

## 3. Discussion

During *L. major* infection, myeloid cells, including monocytes, are recruited to dermal lesions, where they experience hypoxic conditions. Our data show that the longevity of myeloid cells in the inflamed tissue enhances their hypoxic state. We find that the hypoxic environment in leishmanial lesions is associated with the activation of HIF-α signaling. Given that monocytes differentiate into macrophages at the site of infection, and HIF-α activation can impact macrophage function, we investigated the role of HIF-α signaling using a combination of strategies to augment and delete HIF-α activation during *L. major* infection. However, a caveat to our approach is the use of a high-dose model of *Leishmania* infection, which may induce more tissue damage and hypoxia than a more physiological low dose of parasites; therefore, the relevance of hypoxia and HIF-α during a low-dose infection needs to be confirmed. Upon infection, macrophages become activated to phagocytize and kill parasites by NO and ROS. However, macrophages also serve as the host cell and replicative niche for parasites. Macrophages also orchestrate lymphatic remodeling for lesion resolution. Therefore, macrophages play multiple roles in both parasite control and persistence as well as wound healing in CL. Altogether, we found that pharmacological HIF-α activation promoted the ability of macrophages to drive lymphatic remodeling through VEGF-A production during infection, but HIF-α activation did not impact parasite phagocytosis or killing. Alternatively, basal HIF-α signaling restricted macrophage parasite phagocytosis. While HIF-α stabilization did not enhance parasite killing, our findings suggest that pharmacological activation of HIF-α could induce VEGF-A, which would be beneficial for promoting lymphangiogenesis to improve lesion resolution during CL.

Hypoxia promotes macrophage phagocytosis in a HIF-1α-dependent manner, while HIF-2α is not involved in phagocytosis under hypoxic conditions [22]. In this study, we detected higher numbers of internalized parasites in macrophages deficient in myeloid ARNT/HIF-α signaling compared to macrophages with intact ARNT/HIF-α signaling in normoxic conditions. These data suggest that HIF-α inhibits parasite phagocytosis. While HIF-2α may not participate in phagocytosis under hypoxic conditions, basal HIF-2α is necessary and sufficient to suppress phagocytosis and efferocytosis under normoxic conditions [23]. The uptake of apoptotic cells or *Staphylococcus aureus* is higher in LysM^Cre^HIF-2α^f/f^ macrophages compared to controls [23]. Moreover, LysM^Cre^ARNT^f/f^ macrophages (missing both HIF-1α and HIF-2α signaling, such as the macrophages used in our experiments) show enhanced phagocytosis and efferocytosis compared to control macrophages [23]. As a result, we hypothesize that the absence of HIF-2α in our LysM^Cre^ARNT^f/f^ macrophages is responsible for the enhanced parasite uptake during *L. major* infection. Therefore, we propose that HIF-2α acts as a phagocytic repressor during *Leishmania* infection.

HIF-α stabilization leads to lower bacterial and fungal burdens within macrophages [24,25]. Complementing the work of others, our data show that inflammatory stabilization of HIF-α by LPS and IFNγ leads to lower parasite numbers, but chemical HIF-α stabilization by DMOG treatment does not lead to lower parasite numbers [6,9]. HIF-α stabilization can occur by two major mechanisms, including inhibiting prolyl hydroxylase (PHD) enzymes or factor inhibiting HIF (FIH) [26]. While previous studies stabilized HIF-α by inhibiting PHD enzymes without affecting FIH [6], we stabilized HIF-α by DMOG treatment, which inhibits both PHD enzymes and FIH, and showed that chemical HIF-α stabilization alone does not lead to lower parasite numbers in the absence of additional proinflammatory stimuli. Therefore, our body of work continues to support the hypothesis that multiple factors from the tissue microenvironment, such as hypoxia and proinflammatory cytokines, contribute to parasite control. However, the requirement for HIF-α signaling in parasite control is not completely clear. While it has been reported that myeloid HIF-1α is required for the robust killing of *L. major* parasites in a NO-dependent manner, we found that ARNT/HIF-α signaling only plays a minor, although significant, role in parasite control. Given that LysM^Cre^ARNT^f/f^ macrophages could efficiently control parasites in response to LPS and IFNγ, our data suggest that *L. major* parasites are controlled in an ARNT/HIF-α-independent mechanism in vitro. Mice deficient in myeloid HIF-1α exhibited higher parasite numbers following *L. major* infection in vivo, but we did not detect differences in parasite burdens in infected LysM^Cre^ARNT^f/f^ mice, which are deficient in both myeloid HIF-1α and HIF-2α signaling [6,9]. Our in vitro findings here showing that LysM^Cre^ARNT^f/f^ macrophages can control parasites in response to proinflammatory stimuli are consistent with our findings in vivo showing that LysM^Cre^ARNT^f/f^ mice can control parasites at similar levels to LysM^Cre^ARNT^f/+^ controls [8,9]. It should be noted that although LysM^Cre^ARNT^f/f^ macrophages exhibited a slightly higher number of parasites at 72 h following LPS and IFNγ treatment compared to LysM^Cre^ARNT^f/+^ macrophages, suggesting that HIF-α contributes to parasite control, LysM^Cre^ARNT^f/f^ macrophages also internalized more parasites at 2 h, which could be responsible for the slightly higher numbers of parasites in LysM^Cre^ARNT^f/f^ macrophages at 72 h, simultaneously refuting a role for HIF-α-signaling in parasite control. Altogether, these data suggest that parasite control occurs predominantly through a HIF-α-independent mechanism in vitro.

While it is clear that HIF-α subunits are active during infection, the specific factors responsible for HIF-α activation have not been defined and appear to be context-dependent. Hypoxia is hypothesized to be a major driver of HIF-α activation, but proinflammatory signals can also activate HIF-1α and HIF-2α [27,28]. Here, we used VEGF-A production as a surrogate for HIF-α activation during infection. *L. major* parasites can activate TLRs, and TLR activation can lead to HIF-α activation [29,30,31,32,33]. Similarly, *L. major* induces reactive oxygen species (ROS) production, and ROS can activate HIF-α [34,35,36,37], but parasites alone did not induce VEGF-A production by macrophages. These data suggest that parasites, parasite TLR ligation, and parasite-induced ROS production do not contribute to HIF-α activation during infection in vitro. A caveat to our study is that all experiments were performed under normoxic conditions, as we attempted to uncouple hypoxia from HIF-α activation, but it will be important to determine if similar results occur under hypoxic conditions. One might speculate that hypoxia along with DMOG treatment or the addition of proinflammatory stimuli might intensify the expression of HIF-α target genes or VEGF-A production. We tested LPS and IFNγ alone and in combination, given their known roles in stimulating proinflammatory M1 macrophages. We also predict that macrophages are exposed to these molecules during *L. major* infection. LPS may activate skin macrophages during CL, given the presence of the microbiome and the compromised integrity of the skin [38]. IFNγ is also elevated at the site of infection due to the host Th1 immune response [3]. While proinflammatory stimuli have been hypothesized to induce HIF-α activation, we did not find these factors to be robust drivers of macrophage VEGF-A production. However, other inflammatory stimuli such as TLR ligands, e.g., lipoteichoic acids and peptidoglycans from Gram-positive bacteria that are known to dominate in the skin microbiome, should be tested for their ability to induce HIF-α activation and/or VEGF-A production. Alternatively, hypoxia or other soluble mediators in the inflamed skin may drive HIF-α activation and VEGF-A production by macrophages during *L. major* infection. VEGF-A can be induced by a variety of growth factors and cytokines, including FGF2, PDGF, TGF-β, IL-1β, IL-6, IL-8/CXCL8, and TNFα, and some of these soluble mediators act synergistically with hypoxia [39,40,41,42,43,44,45,46]. Importantly, many of these factors are present in leishmanial lesions and may induce VEGF-A during *L. major* infection. As a result, the identification of the factors responsible for VEGF-A production by macrophages is the focus of ongoing investigation in the lab, given the critical role of VEGF-A in lesion resolution during CL.

## 4. Materials and Methods

### 4.1. Mice

Female and male animals used in these experiments were either purchased from the National Cancer Institute or bred in a vivarium on campus. CD45.2^+^ C57BL/6 and CD45.1^+^ C57BL/6 mice were purchased from the National Cancer Institute. Mice with a myeloid-specific *ARNT* conditional knockout were bred by crossing a strain expressing the LysM^Cre^ allele with a strain with a floxed *ARNT* conditional allele [47,48]. The LysM^Cre^ARNT^f/f^ and LysM^Cre^ARNT^f/+^ control mice were a gift from M. Celeste Simon (University of Pennsylvania, Philadelphia, PA). LysM^Cre^ARNT^f/f^ mice were infected alongside LysM^Cre^ARNT^f/+^ controls for experiments. All mice were housed in vivariums under pathogen-free conditions at the University of Arkansas for Medical Sciences (UAMS). All infections were induced in mice between 6 and 8 weeks of age. All procedures performed were approved by UAMS IACUC and followed institutional guidelines.

### 4.2. Parasites

*Leishmania major* and DsRed *L. major* Friedlin strain parasites were grown in vitro in Schneider’s insect media (Gibco) supplemented with 20% heat-inactivated fetal bovine serum (FBS, Invitrogen), 100 U/mL penicillin, 100 U/mL streptomycin, and 2 mM L-glutamine (MilliporeSigma). Metacyclic promastigotes used for infections were isolated from 4–5-day-old cultures using Ficoll gradient separation (MilliporeSigma) [49].

### 4.3. In Vivo Infections

Infections were performed by injecting 2 × 10^6^ parasites in 10 μL of PBS intradermally into the right ear of mice. Mice were anesthetized with ketamine and xylazine prior to infection. Lesion development was monitored weekly by measuring ear thickness, lesion diameter, and pathology to calculate lesion volume. Ears were digested for 90 min at 37 °C with 0.25 mg/mL liberase TL (Roche) with 10 μg/mL DNase I (Sigma) in RPMI 1640 media (Gibco).

### 4.4. Isolation of CD11b^+^ Cells and Adoptive Transfer

Spleens, femurs, tibias, and fibulas were taken from C57BL/6 mice to obtain CD11b^+^ cells. Single-cell suspensions were enriched using a CD11b MicroBead isolation kit (Miltenyi Biotec) to obtain myeloid cells. For positive selection, the autoMACS Pro Separator (Miltenyi) was used. Pooled CD11b^+^ cells from the bone marrow and spleens (purity > 97%) were resuspended in PBS, and 4 × 10^6^ cells were injected into the retroorbital sinus of the recipient C57BL/6 mouse.

### 4.5. Flow Cytometry

Surface staining was performed on dermal cells from ears after enzymatic digestion and processing. To exclude dead cells, cell suspensions were first incubated with LIVE/DEAD Fixable Aqua Dead cell dye (Invitrogen) for 10 min at room temperature. FcγRs were blocked with 2.4G2 anti-mouse CD16/32 antibody (BioXCell) and 0.2% normal rat IgG (BioXCell) for 10 min at 4 °C. For surface staining, cells were stained for 30 min at 4 °C using antibodies: anti-CD45 AF 700 (clone 30-F11), anti-CD45.1 eFlour450 (clone A20), anti-CD45.2 AF 700 (clone 104), anti-Ly6C PerCpCy5.5 (clone HK1.4) (all from eBiosciences); anti-CD11b BV605 (clone M1/70), anti-Ly6G APC (clone 1A8), and anti-CD64 PECy7 (clone X54-5/7.1) (all from Biolegend) in the presence of Brilliant Violet Buffer (BD Biosciences) or Super Bright staining buffer (eBiosciences). Cell events were acquired using the LSRII Fortessa flow cytometer (BD Biosciences) and analyzed using FlowJo software version 10 (Tree Star).

### 4.6. Pimonidazole

Each mouse was injected with 1.5 mg of pimonidazole (Hypoxyprobe kit) in 200 μL of PBS intraperitoneally (i.p.) 90 min before sacrifice to measure hypoxia at the cellular level. Cells were fixed and permeabilized using the Foxp3 intracellular staining kit (eBiosciences) after cell-surface staining. Intracellular staining was carried out with α-pimonidazole-FITC (1:100) according to the manufacturer’s instructions.

### 4.7. mRNA Extraction and Real-Time PCR

mRNA was extracted using the EZNA Total RNA Kit I (Omega BioTek) and reverse transcribed using high-capacity cDNA Reverse Transcription (Applied Biosystems). Quantitative real-time PCR (qPCR) was performed on a QuantStudio 6 Flex Real-Time PCR system (Life Technologies) with SYBR Green PCR Master Mix. qPCR results were normalized to the housekeeping gene ribosomal protein S11 (RPS11) with a comparative threshold cycling method (2^−ΔΔCT^) to quantify. The following mouse primers were selected from Harvard’s Primer Bank (https://pga.mgh.harvard.edu/primerbank/, accessed on 4 October 2021): *Vegfa* (Forward 5′-ATCTTCAAGCCGTCCTGTGT-3′ and Reverse 5′-GCATTCACATCTGCTGTGCT-3′), *Nos2* (Forward 5′-ATGGAGACTGTCCCAGCAAT-3′ and Reverse 5′-GGCGCAGAACTGAGGGTA-3′), *Epo* (Forward 5′-CATCTGCGACAGTCGAGTTCTG-3′and Reverse 5′-CACAACCCATCGTGACATTTTC-3′), *Pgk1* (Forward 5′-ATGTCGCTTAACAAGCTG-3′ and Reverse 5′-GCTCCATTGTCCAAGCAGAAT-3′), *Arg1* (Forward 5′-CTCCAAGCCAAAGTCCTTAGAG-3′ and Reverse 5′-AGGAGCTGTCATTAGGGACATC-3′), and *Rps11* (Forward 5′-CGTGACGAAGATGAAGATGC-3′ and Reverse 5′-GCACATTGAATCGCACAGTC-3′).

### 4.8. RNA Sequencing (RNA-Seq): Data Analysis

Following demultipexing, RNA reads were surveyed for sequencing quality using FastQC (version 1.7) (http://www.bioinformatics.babraham.ac.uk/projects/fastqc, accessed on 4 October 2021) and MultiQC (version 1.6) [50]. Next, the raw reads were processed according to Lexogen’s QuantSeq data analysis pipeline with slight modification. Residual 3′ adapters, polyA read-through sequences, and low-quality (Q < 20) bases were trimmed using BBTools BBDuk (version 38.52) (https://sourceforge.net/projects/bbmap/, accessed on 4 October 2021). Additionally, the first 12 bases were also removed per the manufacturer’s suggestion. Cleaned reads (≥20 bp) were mapped to the mouse reference genome (GRCm38/mm10/ensemble release-84.38/GCA_000001635.6) using STAR (version 2.6.1a), permitting up to 2 mismatches depending on the alignment length (i.e., 0 mismatches for 20–29 bp; 1 mismatch for 30-50 bp; 2 mismatches for 50-60+ bp) [51]. Reads mapping to >20 locations were discarded. Gene level counts were quantified using HTSeq (htseq-counts) (version 0.9.1) (mode:intersection-nonempty) [52]. Genes with unique Entrez IDs and at least ~2 counts per million (CPM) in 4 or more samples were chosen for statistical testing. Next, scaling normalization using the trimmed mean of M-values (TMM) method was used to correct for compositional differences between sample libraries [53]. Differential expression was assessed using limma voomWithQualityWeights function with empirical Bayes smoothing [54]. Genes with Benjamini–Hochberg adjusted *p*-values < 0.05 and fold-changes > 2 were considered significant [55]. Gene set enrichment analysis (GSEA) using Kyoto Encyclopedia of Genes and Genomes (KEGG) signaling pathways was carried out using EGSEA with default parameters [56].

### 4.9. Generation of Bone Marrow-Derived Macrophages

Femurs were removed from mice and soaked in 70% ethanol for 2 min and then flushed with 10 mL of cDMEM. The cells were resuspended, counted, and plated at 5 × 10^6^/mL in a 100 mm Petri dish in 10 mL of conditioned macrophage media (cDMEM with 25% of L929 cell supernatants) for 7 days. At day 3 of 7, an additional 10 mL of conditioned macrophage media was added to each Petri dish. At day 7, the cells were washed with ice-cold PBS and harvested by dislodging by pipetting and a cell scraper. The cells were then pooled and counted to plate into either 24-well plates (1 × 10^6^ cells in 1 mL) or 48-well plates (5 × 10^5^ cells in 500 µL).

### 4.10. In Vitro Infections with DMOG Treatment

Bone marrow-derived macrophages (BMDMs) were plated as described above into 24-well plates overnight. Macrophages were cultured with *L. major* parasites (MOI 5:1), and extracellular parasites were washed away after 2 h. Following the washes, cells were cultured in triplicate with 100 ng/mL LPS (Sigma), 10 ng/mL IFNγ (Peprotech), with or without 0.1, 0.2, or 0.3 mM DMOG (Sigma) for 2, 24, or 72 h. Pretreated macrophages were incubated overnight with DMOG prior to infection.

### 4.11. Microscopy for Parasite Quantification In Vitro

After infection with DsRed *L. major* parasites, BMDMs were stained by removing conditioned media from each well and washing with PBS before fixation by methanol at -20 °C for 3 min. After fixation, the methanol was removed, and the wells were washed twice with PBS before staining with DAPI in PBS (Invitrogen) for 5 min in the dark at room temperature. After staining, the DAPI solution was removed, and the wells were again washed twice with PBS. After washing, 500 µL of PBS was added to each well to keep the cells hydrated. The plate was then wrapped in aluminum foil and stored in a 4 °C refrigerator. Fluorescence imaging was performed on a Keyence BZ-X810 using the 20X Plan-Fluor NA 0.45 objective in high-resolution mode. Five images were taken at random locations in each well in both the DAPI and DsRed/TxRed channels. Cell counts were recorded using the BZ Image Analyzer (Keyence). Parasite counts were normalized to macrophage number by dividing the average number of parasites by the average number of macrophages for an individual well.

### 4.12. VEGF-A Production

Cell-free supernatants were collected at 24 h following parasite infection to measure VEGF-A production using the Mouse VEGF-A ELISA kit according to the manufacturer’s instructions (R and D Systems).

### 4.13. Statistics

All data were analyzed using GraphPad Prism 8 or 9, and *p* ≤ 0.05 was considered statistically significant. Statistical significance was calculated using a 2-tailed Student’s unpaired or paired *t*-test for a single comparison between groups. Grubbs’ test was used to identify and mathematically remove outlier data points. For multiple-comparison analysis, statistical significance was determined by a one-way analysis of variance (ANOVA) followed by the post hoc Tukey’s test with no designated control group or a Mann–Whitney test with a designated control group.

## Figures and Tables

**Figure 1 pathogens-10-01584-f001:**
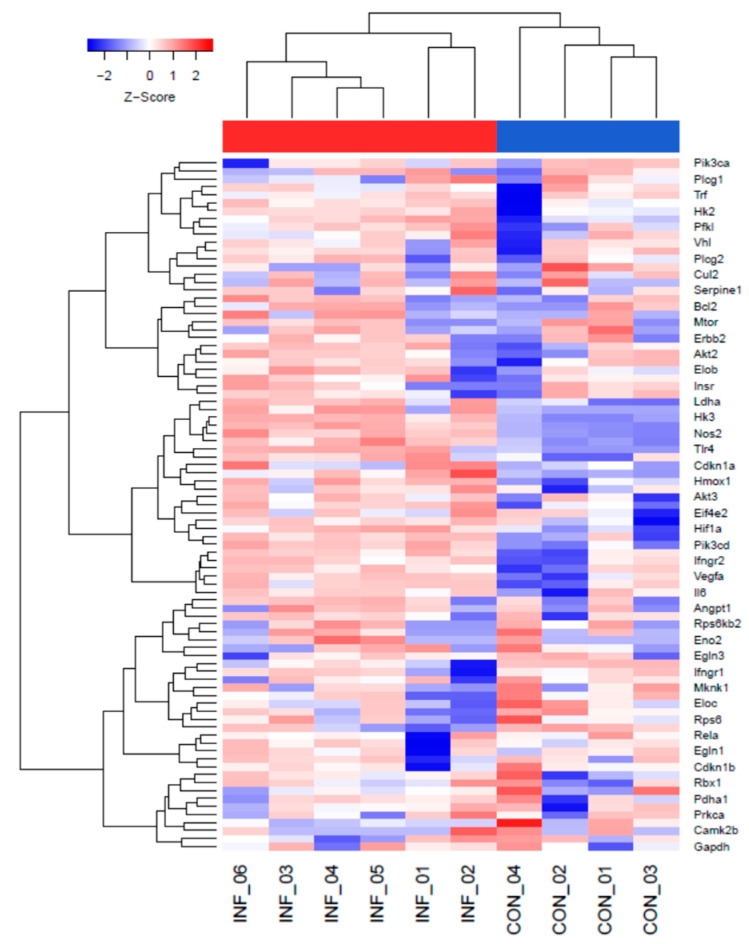
The HIF-1α signaling pathway is elevated following infection with *L. major* parasites in vivo. C57BL/6 mice were infected with 2 × 10^6^
*L. major* metacyclic promastigote parasites intradermally in the ear. At 4 weeks p.i., naïve control (N = 4) and infected (N = 6) ears were subjected to RNA-Seq analysis. Hierarchical clustering of the expression profile was performed. Heat maps indicate fold changes in gene expression in *L. major*-infected ears with >2-fold (red) or <2-fold (blue) compared to naïve controls. The HIF-1α signaling pathway was obtained using the KEGG pathway analysis database. Relative expression was normalized with the Z score.

**Figure 2 pathogens-10-01584-f002:**
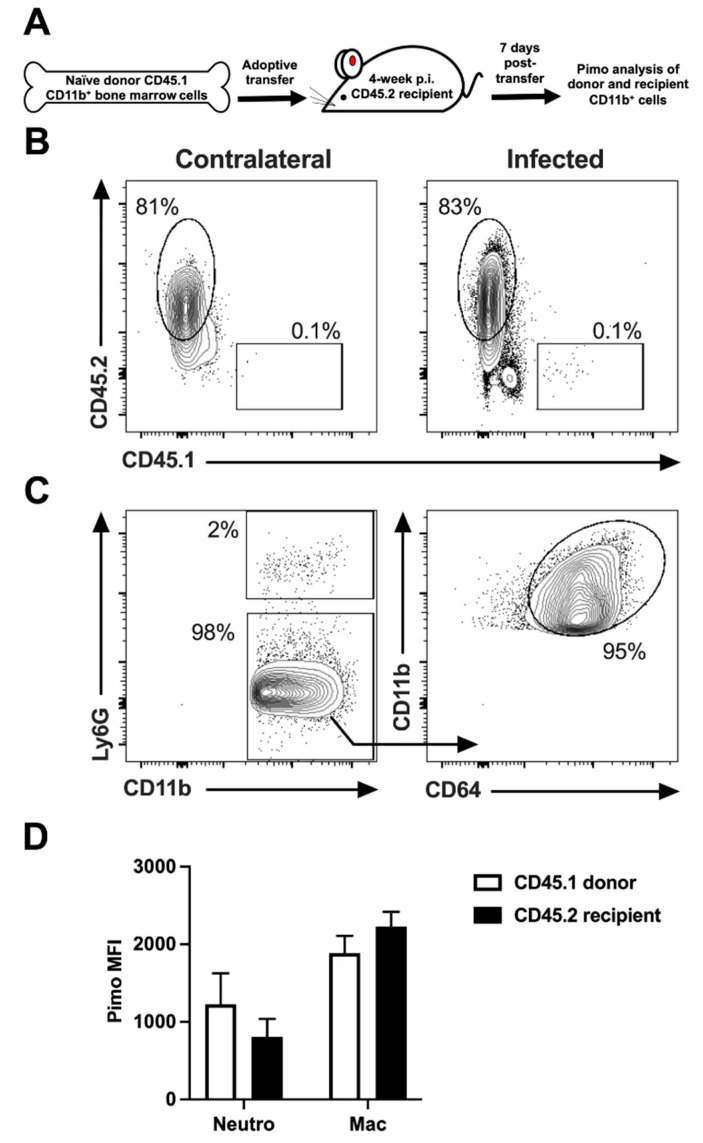
Myeloid cells experience inflammatory hypoxia as they enter the dermal tissue during *L. major* infection. C57BL/6 mice were infected with *L. major* parasites intradermally in the ear. At 4 weeks p.i., CD45.1^+^CD11b^+^ cells isolated from the bone marrow and spleens of naïve mice were adoptively transferred retroorbitally into CD45.2^+^ infected mice. At 7 days post-transfer, infected mice were given pimonidazole i.p. 90 min before euthanasia, and cells from infected ears were stained for pimonidazole and analyzed by flow cytometry. (**A**) Illustrated model of CD11b^+^ adoptive transfer experiment. (**B**) Representative dot plots showing that CD45.1^+^ transferred cells are present in infected ears of CD45.2^+^ recipient mice. Cells were previously gated on total, live singlet events. (**C**) Representative dot plots showing flow cytometry gating strategy for CD11b^+^Ly6G^+^ neutrophils and CD11b^+^Ly6G^−^CD64^+^ macrophages. (**D**) Quantification of pimonidazole (pimo) median fluorescence intensity (MFI) after gating on the myeloid cells: CD11b^+^Ly6G^+^ neutrophils and CD11b^+^Ly6G^−^CD64^+^ macrophages. Data shown here are from one experiment and representative of two independent experiments with 5 mice per group. Data are presented as the mean +SEM with no significant differences in pimonidazole MFIs between donor cells and endogenous host cells of the same cell type as analyzed by a paired *t*-test.

**Figure 3 pathogens-10-01584-f003:**
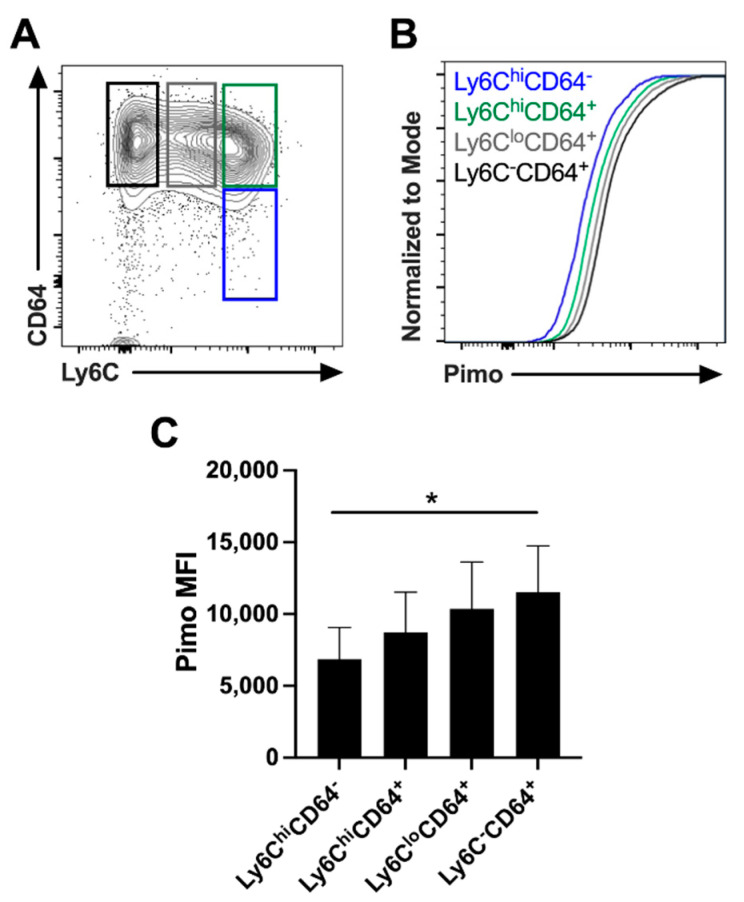
Myeloid cell longevity in the dermal tissue is associated with inflammatory hypoxia during *L. major* infection. C57BL/6 mice were infected with *L. major* parasites intradermally in the ear. At 5 weeks p.i., mice were given pimonidazole i.p. 90 min before euthanasia, and infected ears were stained for pimonidazole and analyzed by flow cytometry. (**A**) Representative flow cytometry contour plot showing subpopulations in the monocyte-to-macrophage transition from Ly6C^hi^CD64^−^ inflammatory monocytes to Ly6C^−^CD64^+^ macrophages after gating on total, live, singlet, and CD45^+^CD11b^+^Ly6G^−^ cells. (**B**) Representative flow cytometry cumulative distribution function (CDF) plot showing pimonidazole MFI after gating on the myeloid Ly6C^hi^CD64^−^, Ly6C^hi^CD64^+^, Ly6C^lo^CD64^+^, and Ly6C^−^CD64^+^ subpopulations in Figure (**A**). (**C**) Quantification of pimonidazole MFI in myeloid subpopulations from infected skin from Figure (**B**). Data shown here are from one experiment and representative of two independent experiments with 5 mice per group. Data are presented as the mean +SEM. * *p* < 0.05, one-way ANOVA followed by the post hoc Tukey’s test comparing pimonidazole MFI between all subpopulations.

**Figure 4 pathogens-10-01584-f004:**
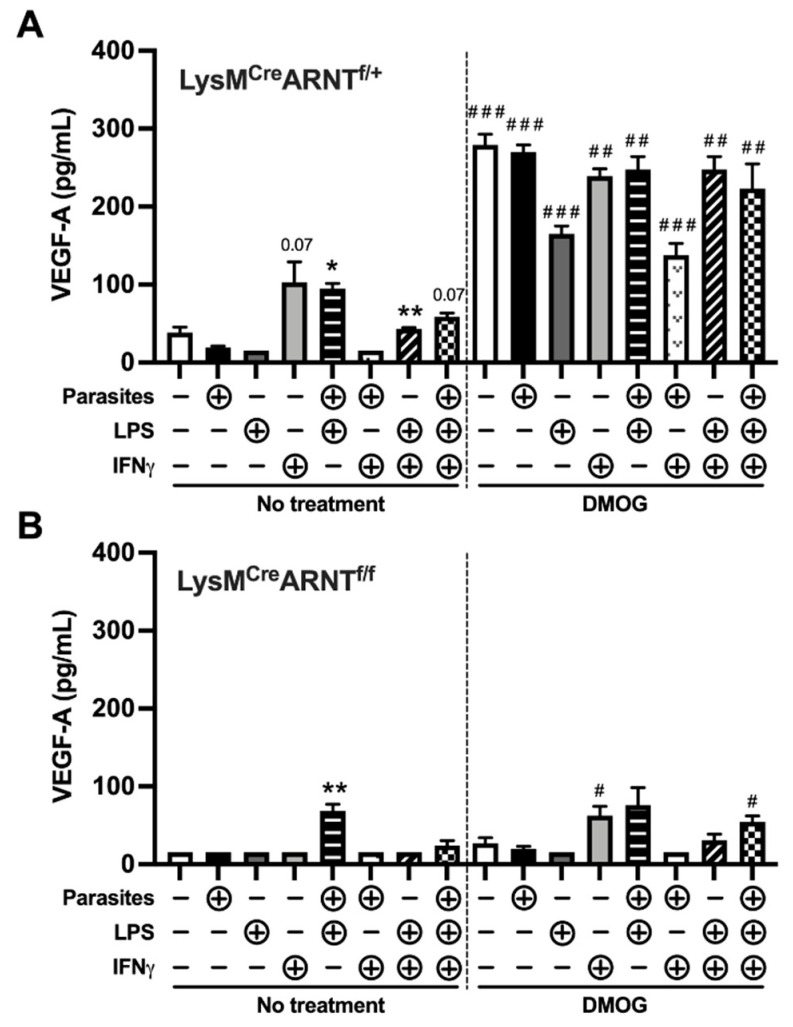
Macrophages produce VEGF-A in response to proinflammatory stimuli and HIF-α stabilization during *L. major* infection in an ARNT/HIF-dependent manner. BMDMs were cultured with LPS (100 ng/mL) alone, IFNγ (10 ng/mL) alone, or both LPS (100 ng/mL) and IFNγ (10 ng/mL) in macrophages that were infected (or not) with *L.*
*major* parasites (5:1 MOI). BMDMs were also cultured under these conditions with and without 0.2 mM DMOG, which stabilizes HIF-α. BMDMs were generated from (**A**) LysM^Cre^ARNT^f/+^ or (**B**) LysM^Cre^ARNT^f/f^ to determine the contribution of ARNT/HIF signaling for VEGF-A production. Supernatants were collected after 24 h. VEGF-A production was quantified by ELISA. Data are presented as mean +SEM. ** *p* < 0.01, * *p* < 0.05, t-test comparing proinflammatory stimuli to media alone; ### *p* < 0.005, ## *p* < 0.01, # *p* < 0.05, *t*-test comparing the same treatment condition with and without DMOG. Although not depicted in the figure, it should be noted that all LysM^Cre^ARNT^f/+^ BMDMs treated with DMOG produced significantly more VEGF-A than LysM^Cre^ARNT^f/f^ BMDMs treated with DMOG (*p* < 0.01 by *t*-test comparing different mouse strains exposed to the same condition).

**Figure 5 pathogens-10-01584-f005:**
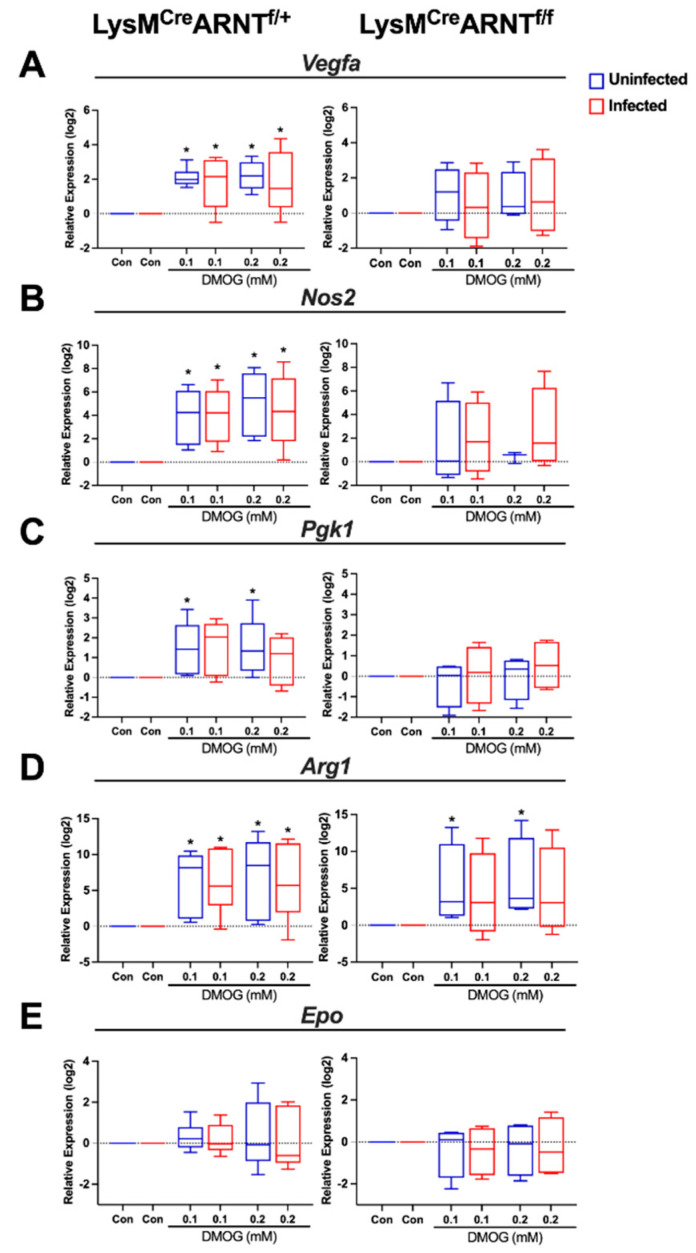
DMOG treatment induces HIF-1α and HIF-2α activation during *L. major* infection. BMDMs were generated from LysM^Cre^ARNT^f/+^ or LysM^Cre^ARNT^f/f^ mice. Macrophages were infected (or not) with *L. major* parasites (5:1 MOI) and cultured with and without 0.1 mM or 0.2 mM DMOG for 24 h. The expression of *Vegfa* (**A**), *Nos2* (**B**), *Pgk1* (**C**), *Arg1* (**D**), and *Epo* (**E**) was analyzed by quantitative real-time PCR. Relative mRNA expression was normalized to the housekeeping gene *RpsII*. Results shown here are the mean +SEM of the fold change over untreated controls (Con) pooled from 4–6 individual experiments. * *p* ≤ 0.05, Mann–Whitney test comparing infected DMOG-treated to infected untreated controls (blue to blue) or comparing uninfected DMOG-treated to uninfected untreated controls (orange to orange).

**Figure 6 pathogens-10-01584-f006:**
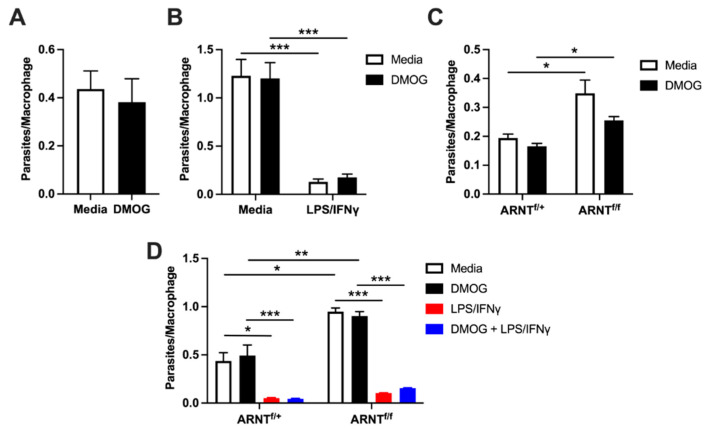
HIF-α stabilization does not impact parasite burdens in macrophages following *L. major* infection. For all experiments, the number of parasites per macrophage was quantified by fluorescence microscopy. (**A**) C57BL/6 BMDMs were pretreated overnight with or without 0.2 mM DMOG before infection with fluorescently labeled DsRed *L. major* parasites (MOI 5:1). Parasites were quantified at 2 h p.i. (**B**) C57BL/6 BMDMs were infected with DsRed *L. major* and cultured for 72 h with or without 0.2 mM DMOG in the presence or absence of 100 ng/mL LPS + 10 ng/mL IFNγ. (**C**) BMDMs derived from LysM^Cre^ARNT^f/+^ and LysM^Cre^ARNT^f/f^ mice were pretreated overnight with and without 0.2 mM DMOG before being infected with DsRed *L. major* for 2 h. (**D**) LysM^Cre^ARNT^f/+^ and LysM^Cre^ARNT^f/f^ BMDM were cultured for 72 h after infection with or without 0.2 mM DMOG in the presence or absence of 100 ng/mL LPS + 10 ng/mL IFNγ. Results shown in (**A**,**B**) are pooled from 4 individual experiments. Results shown in (**C**,**D**) are a single experiment representative of 3–4 individual experiments. Data presented as the mean +SEM. * *p* < 0.05, ** *p* < 0.01 and *** *p* < 0.005, *t*-test comparing DMOG and LPS/IFNγ treatment to media alone and conditioned media, or *t*-test comparing LysM^Cre^ARNT^f/+^ and LysM^Cre^ARNT^f/f^ macrophages under the same treatment conditions.

## Data Availability

Data are contained within the article, and the data from our bulk RNA-Seq analysis were deposited in Gene Expression Omnibus (GEO accession number—GSE185253).

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
