# Peer review of "HIF-α Activation Impacts Macrophage Function during Murine Leishmania major Infection"

_pathogens, 2021, doi:10.3390/pathogens10121584_

Round 1

Reviewer 1 Report

The manuscript “HIF-a activation impacts macrophage function during murine Leishmania major infection”  describes and considers the role of HIF during Leishmania infection.  The topic is certainly interesting, although it has already been investigated by other authors in previous years ( for example I. Mesquita et al. 2020 or V. Schatz et al. 2016). Nevertheless, the manuscript is well structured, the experiments conducted investigated the subject matter in a precise and focused manner, The discussions are well articulated and very comprehensive. In my opinion, the manuscript can be accepted for publication.

There are a few observations I would like to make to the authors:

lines 185, 232,234 L. maior in italics

Lines 201, 245, 249 in vivo and in vitro in italics

Reviewer 2 Report

The manuscript by Bettadapura, Weinkopff et al investigates the impact of HIF-a activation on macrophage function in mouse model of cutaneous leishmaniasis. They show that although HIF-a-related transcripts are induced in infected ear dermis, infection with L. major promastigotes alone is not sufficient to induce VEGF-a transcription or secretion by BMDM in vitro, at least not in normoxic conditions. However, concurrent treatment with LPS resulted in enhanced VEGF-a secretion. HIF-a stabilization with DMOG also promoted a transcriptional response, but this response was not further influenced by infection with L. major. Finally the authors show using ARNT-deficient BMDM that basal HIF-a decreases parasite uptake, and propose that this effect is due to the activity of HIF-2a. These results follow the authors’ previous work on the role of ARNT/HIF signaling in regulating cutaneous leishmaniasis resulting from infection with L. major, and bring some new mechanistic insight as to the potential importance of inflammatory signals from the environment in HIF-a stabilization. However, there are a few caveats as detailed below that should be addressed in order to strengthen the authors’ conclusions.   

  1. The number of parasites used for the intradermal infection is very high, especially if the authors are indeed using metacyclic promastigotes. Such a high dose enhances tissue damage and is likely to create artefacts that would not occur at a more physiological dose. The authors should discuss this.
  2. Why the pimonidazole staining is not shown in Figure 2? Overall, although the contralateral ear is likely to recruit fewer monocytes, the best staining control would be an infected ear from a mouse that has not received CD45.1+ cells due to potential differences in background autofluorescence induced by the infection. As is, the percentage of recruitment (CD45.1+ cells) is not very convincing, which makes the authors’ interpretation of their data (similar levels of hypoxia in monocyte-derived and resident/endogenous macrophages) less solid. If spleen cells were included (as per materials and methods), why refer to “bone marrow” cells rather than CD11b+ cells? Furthermore, 1 week post-transfer is likely to be too late for analysis, especially of mature neutrophils that have a short half-life.
  3. In Figure 4, do all conditions promote infection/phagocytosis to a similar degree? Parasite + IFN appears to be associated with decreased VEGF-a secretion, independent of ARNT. Is this due to decreased parasite uptake and/or viability or to other IFN-dependent effects? Why was LPS specifically chosen to stimulate inflammation in this context (relative importance of Gram-negative vs -positive bacteria in skin microbiome)? Would other TLR ligands (such as peptidoglycan) yield similar results?
  4. All in vitro experiments were performed in normoxic conditions. How may this influence the results? This should be discussed.

Specific comments

Lines 173-179 – please reformulate. There seems to be some repetition here.

Figure 3 – The combination of fluorochromes (Ly6C PerCP-Cy5.5 and pimonidazole-FITC) could be an issue here. The differences are very subtle and it would be important to validate that the result is not an artefact from compensation (inverse correlation with Ly6C levels). The authors should show a similar analysis for a negative control (all antibodies without pimonidazole injection). If real, are such small differences likely to be biologically significant?

Lines 260-268 – please reformulate. There seems to be some repetition here.

How efficient is LysMCre-mediated gene deletion in BMDM? Could mosaic deletion explain some of the (considerable) variability in the results (for example, in Figure 5)?

The overall importance of Figure 5 is not clear. There does not seem to be any difference between infected and uninfected samples

Lines 375-377 – The authors show that LPS/IFN-treatment leads to reduced parasite numbers, but their data do not permit to conclude that the treatment promotes parasite killing in a HIF-dependent manner. There may be a slight difference between ARNT-f/f and ARNT-f/+ conditions, but ARNT-f/f macrophages already show increased parasite uptake at 2h. It is therefore impossible to conclude that the difference in parasite numbers at 72h is due to HIF-dependent killing.

Round 2

Reviewer 2 Report

I am satisfied with the authors' reply and modifications made to the text.